

# Role of Atmospheric Circulations on Haze Pollution in December 2016

Zhicong Yin [12] and Huijun Wang [123]

[1]Key Laboratory of Meteorological Disaster, Ministry of Education / Joint International Research Laboratory of Climate and
Environment Change (ILCEC) / Collaborative Innovation Center on Forecast and Evaluation of Meteorological Disasters
(CIC-FEMD), Nanjing University of Information Science & Technology, Nanjing 210044, China

[2]Nansen-Zhu International Research Centre, Institute of Atmospheric Physics, Chinese Academy of Sciences, Beijing, China

[3]Climate Change Research Center, Chinese Academy of Sciences, Beijing, China

*Correspondence to*: Zhicong Yin (yinzhc@163.com)

**Abstract.** In the east of China, recent haze pollution has been severe and damaging. In addition to anthropogenic emissions, atmospheric circulations and local meteorological conditions were conductive factors. The number of December haze days over North China and the Huanghuai area has increased sharply since 2010 and was greatest in 2016. During 2016, the most aggressive control measures for anthropogenic emissions were executed from 16–21 December, but the most severe haze pollution still occurred, covering approximately 25% of the land area of China and lasting for 6 days. The atmospheric circulations must play critical roles. The associated atmospheric circulations that were verified by climatic correlation analysis were a weaker East Asia jet stream in the upper troposphere, a positive phase of the East Atlantic/West Russia pattern in the middle troposphere and conductive local weather conditions (lower PBL, small surface wind speed, and abundant moisture) near the surface. The influence of sea surface temperature near the Gulf of Alaska and the subtropical eastern Pacific, snow cover in western Siberia and associated physical processes on haze pollution are also discussed.





## 1. Introduction

Because of its enormous adverse effects, haze pollution has become one of the most serious environmental problems in China, attracting considerable scientific and social attention. Increasing anthropogenic emissions have been contributing to severe haze pollution in China and mainly impacted on the long-term trend of haze days (Wang et al. 2013). However, interannual variations of haze days were affected by meteorological conditions (Wang et al. 2015; Yang et al. 2016; Wang and Chen. 2016). At present, aerosols have approached saturation in the atmosphere (Zhang et al. 2013). When the horizontal and vertical dispersion of atmospheric particulates is impeded, haze weather tends to occur (Yin et al. 2015a). Therefore, anomalous atmospheric circulations play a key role in the formation of heavy haze pollution in winter (December–February) (Chen and Wang 2015). From the hemispheric and regional perspective, the positive phase of the Arctic Oscillation (Yin et al. 2015b), the weak East Asia winter monsoon (Li et al. 2015; Yin et al. 2015b) and the positive phase of the East Atlantic/West Russia (EA/WR) teleconnection (Yin et al. 2017) contribute to the occurrence of winter haze by modulating local anticyclone anomalies over North China. As a key local circulation, this anomalous anticyclone resulted in descending motion (Wu et al. 2017) that contributed to a reduction in the height of the planetary boundary layer (PBL). Other conductive weather conditions include reduced surface wind speed and enhanced humidity in the lower atmosphere (Ding et al. 2014). Such weather conditions trap abundant atmospheric particles and moisture, leading to a high concentration of pollutants. Furthermore, the frequency and persistence of weather conditions conducive to the Beijing winter severe haze events were projected to increase substantially under climate change in the future (Cai et al. 2017).

On the sub-seasonal time scale, haze pollution in December is quite serious and has distinct characteristics, but it has not acquired adequate attention. As shown in Figure 1, there have been eight wide scale haze pollution events in China in 2016. During six of these events, the highest $PM_{2.5}$ concentration was observed in North China. During 2016, haze pollution was most severe during 16–21 December with the highest $PM_{2.5}$ concentration of 1100 μg/m$^3$ in the whole of North China and the Huanghuai area (NH, located at 30–41$^o$N, 110–120$^o$E) where more than 300 million people live. The affected area was 2680 thousand km$^2$. Thereinto, the area affected by severe haze was 710 thousand km$^2$, which was close to the total area of the preceding 7 episodes in 2016. In addition, its duration was 6 days, which was approximately twice as long as the other haze episodes (Figure 1b). Furthermore, for the past 38 years, the number of December haze days (DHD) over the NH area ($DHD_{NH}$) was greatest in 2016 (Figure 1a). Since 2010, $DHD_{NH}$ has experienced a sharp increase and reached 21.5 in 2016,



meaning that the air was polluted for approximately 70% of the days. Because air pollution is regulated and controlled by the Chinese Government, annual pollutant emissions varied slowly (Mathews and Tan, 2015), but this could not fully explain the sharp increase of DHD$_{NH}$ after 2010. In particular, although vehicle control and production restriction measures were timely, extensive and strictly implemented, haze pollution was still severe during 16–21 December. The effects of emissions

reduction measures on air pollutants were efficient and proven during the 2015 World Championships and Parade (WCP) held in Beijing (Zhou et al 2017). Thus, understanding the role of atmospheric circulation on extreme haze pollution in December 2016 is vital, and this is analyzed in this paper.

The data and methods are described in section 2. In section 3, we analyzed the roles of global and regional atmospheric circulations on haze in December 2016. Then, a synoptic case (i.e., the severest haze pollution in 2016) was studied to

understand the physical mechanisms in more detail in section 4. A discussion of our results and the main conclusions of the study are included in section 5.

## 2. Datasets and methods

The geopotential height at 500 hPa (Z500), zonal wind at 200 hPa (U200), wind at 850 hPa, surface, sea level pressure (SLP), surface air temperature (SAT), surface lift index, surface relative humidity and vertical wind (omega) were available on the

website of the National Centers for Environmental Prediction/National Center for Atmospheric Research (NCEP/NCAR). These NCEP/NCAR reanalysis I datasets had a horizontal resolution of 2.5 °×2.5 ° from 1979 to 2016 (Kalnay et al. 1996). The height of PBL (1°×1 °) was derived from the ERA-Interim dataset (Dee et al. 2011). The monthly mean Extended Reconstructed SST datasets with a horizontal resolution of 2°×2 ° from 1979 to 2016 were obtained from the National Oceanic and Atmospheric Administration (Smith et al. 2008). The monthly 1 ° by 1 ° snow cover data were supported by the

Rutgers University (Robinson et al. 1993). The EA/WR index was computed by the NOAA climate prediction center according to the Rotated Principal Component Analysis used by Barnston et al. (1987). The routine meteorological measurements included relative humidity, visibility and wind speed at surface that were collected eight times per day. The temperature profile was collected with a sounding balloon twice per day. The calculation procedure for the haze data was consistent with that of Yin et al. (2017). Hourly PM$_{2.5}$ concentration data were downloaded from the website of the Ministry

of Environmental Protection of China. Definitions of anomalies are described in the captions for each figure.



### 3. Associated atmospheric circulations in December 2016

Figure 2 shows the distribution of atmospheric circulation anomalies in December 2016. In the upper troposphere, the East Asia jet stream (EAJS) was weaker and northward relative to the mean status, indicating that meridional cold air activity in East Asia was restricted (Chen et al. 2015). As a result, the land surface of East China was warmer (Figure 2b). On the mid-level, there were positive anomalies of Z500 over Europe and North China and negative centers over the Central-North Atlantic and to the north of the Caspian Sea (Figure 2a). This Rossby wave train resembled the positive phase of the EA/WR pattern. To verify the relationship between the $DHD_{NH}$ and EA/WR pattern, the correlation coefficient was calculated; it was 0.66 after removing the linear trend and exceeded the 99% confidence level (Table 1). This positive correlation was stronger than that with winter haze days (i.e., 0.43), as analyzed by Yin et al. (2017). Specific to the positive Z500 anomalies over 105–125$^{o}$E, 30–50$^{o}$E (i.e., the easternmost center of EA/WR pattern), the correlation coefficient was 0.62 (Table 1). Thus, the anomalous anticyclone over the NH area in December 2016 could efficiently weaken the vertical motion, resulting in shallower PBL and weaker surface lift motion (Figure 3a). Furthermore, possible conductive local weather conditions that included shallow PBL (impacting vertical dispersion), small surface wind speed (impacting horizontal dispersion) and high relative humidity (impacting moisture absorption), whose correlation coefficients with $DHD_{NH}$ were –0.59, –0.63 and 0.49, all passed the 99% confidence test (Table 1). Near the surface, the SLP gradient between Eurasia and West Pacific decreased (Figure 2b) and induced southerly over the east of China. Warm and humid airflow from the south caused the surface wind speed to be slower and the surface relative humidity to be higher (Figure 3b). Under control of such atmospheric circulations and local meteorological conditions, the horizontal and vertical dispersion of atmospheric particulates was suppressed and the haze occurred frequently. In addition, high humidity supported a beneficial environment for the hygroscopic growth of haze matters.

### 4. A synoptic case study

On 15 December of 2016, the Ministry of Environmental Protection of China warned that severe haze pollution would occur over the NH area in the coming week. After that, nearly 30 cities were issued an air pollution red (the highest level) warning, and another 20 cities were issued an orange (the second level) warning (figure omitted). There was a haze-prone zone located from southwest to northeast, i.e., from the north of Henan Province to Beijing. In this haze-prone zone, vehicle control and production restriction measures were both strictly implemented. In the surrounding cities, the industrial



production was also restricted (Figure 4a). These regulatory measures were distributed according to the measured PM$_{2.5}$ concentration, illustrating good scientific-based decision-making and management. Anthropogenic emissions were also more stringently limited at this time than for other haze weather processes that had occurred during the same year, but the most

severe haze pollution still occurred. The highest PM$_{2.5}$ concentration (i.e., 1100 μg/m$^3$) was observed in Shijiazhuang, the provincial capital of Hebei Province (Figure 4b). The measured maximum hourly PM$_{2.5}$ concentrations over the NH area were almost above 500 μg/m$^3$, which was beyond the level of severe air pollution for China. Furthermore, there were three groups of stations with PM$_{2.5}$ concentrations greater than 700 μg/m$^3$, and these were in the central Shaanxi Province, the North of Henan Province and south of Hebei Province, and the central Shandong Province. In addition, the coverage for this

haze pollution process was quite large. Spreading to the south, PM$_{2.5}$ concentrations larger than 300 μg/m$^3$ could be observed in most sites in Jiangsu Province. Around the north edge of the haze area, high PM$_{2.5}$ concentrations occurred in Liaoning Province and Inner Mongolian.

Low visibility is another representation of haze that is widely used in meteorology. Area-averaged visibility was lower than 10 km, and haze pollution was gradually aggravated from 16-21 December (Figure 5). During 16–18 December, the diurnal

variation of visibility was obvious, i.e., visibility decreased at night and increased a little in the morning. Then, visibility decreased persistently and to a minimum value on 21 December. The correlation coefficient between visibility and surface wind speed (surface relative humidity) was 0.4 (–0.69), passing the 99.99% confidence test. The continuous small surface wind speed (<2 m/s) restrained the horizontal dispersion of aerosols, and high humidity in the environment promoted hygroscopic growth that dramatically reduced visibility. The intensity of the temperature inversion remained positive for 132

hours and reached 9 ℃ on 20 December, so atmospheric particles were limited to a shallow PBL and accumulated easily. The meteorological conditions also showed obvious diurnal variation during the early stage of this haze process. Relative humidity was continuously above 80% after 20 December, resulting in persistent decreasing and minimum value of visibility.

The anomalies of atmospheric circulation during 16–21 December were similar to those throughout December, but they were more evident. The EAJS was weaker than the mean status and moved northward, resulting in weak cold air activity and a

warmer surface (Figure 6). In the middle troposphere, the EA/WR pattern could be clearly recognized, and the anomalous anticyclone over North China and Japan was very strong. Under their influence, there was a descending motion from 30$^{\mathrm{o}}$N to 55$^{\mathrm{o}}$N (Figure 7), and the height of PBL was approximately 400 m lower than the mean status of December (Figure 8).



Furthermore, the anomalous height was almost negative all month. In addition to vertical accumulation, there was northward and horizontal transportation of atmospheric particles from the surface to 950 hPa (Figure 7). Near the surface, the SLP of the mid-high latitude was distributed as the positive phase of the AO pattern and the cold air of Polar Regions was toward the Aleutian Islands, so the cold air was difficult to move southward to the NH area. The gradient of SLP (SAT) between Eurasia and the west Pacific receded. The stimulated southerly over the east China coastal area not only weakened the surface wind speed but also led to high humidity over the NH area. In summary, during 16–21 December, atmospheric circulations resulted in highly conductive local weather conditions to severe haze pollution over the NH area.

**5. Discussion and Conclusions**

The most forceful controlling measures of anthropogenic emissions in 2016 were executed during 16–21 December, but the severest haze pollution still occurred, covering approximately 25% of the land area of China and lasting for 6 days. The highest $PM_{2.5}$ concentration observed was 1100 μg/m$^3$. Thus, it was hypothesized that atmospheric circulation must play a critical role. Our results verified that a weaker and northward EAJS led to weak cold air activity. In the middle troposphere, the positive phase of the EA/WR pattern was evident, and it stimulated a descending motion from 30$^o$N to 55$^o$N and lower PBL over the NH area. Near the surface, the positive phase of the AO pattern confined the cold air to move southward. The anomalous southerly not only weakened the surface wind speed but also led to high humidity over the NH area. The atmospheric circulations were very conductive to severe haze pollution over the NH area. During all of December, the number of $DHD_{NH}$ increased sharply from 2010 and was greatest in 2016. The associated atmospheric circulations that were verified by climatic correlation analysis were similar. In other words, there was a weaker EAJS in the upper troposphere, a positive phase of EA/WR pattern in the middle troposphere and conductive local weather conditions (lower PBL, small surface wind speed, and abundant moisture).

The preceding autumn SST in the Pacific significantly influenced the winter haze days in North China (Yin et al. 2016) and could partly explain the severe haze pollution during the winter of 2014 (Yin et al. 2017). For December, the significantly correlated SST with $DHD_{NH}$ was located near the Gulf of Alaska and the subtropical eastern Pacific (Figure 10a). The preceding autumn SST of these two areas was averaged as an index ($SST_{EP}$), and the correlation coefficients with December Z500 were calculated and are shown in Figure 10b. The EA/WR pattern, especially the anomalous anticyclone over NH and





Japan, was obvious. The correlation coefficient between $SST_{EP}$ and EA/WR index ($DHD_{NH}$) was 0.48 (0.55) after detrending; thus, we speculated that the $SST_{EP}$ influenced $DHD_{NH}$ by modulating the EA/WR pattern. The positive SST anomalies near

the Gulf of Alaska and the subtropical eastern Pacific could impact the wave activity flux (WAF) and (Figure 10b) and stimulated a Rossby wave-like pattern propagating from eastern Pacific, through North America and Atlantic and to East Asia. The atmospheric action centers over North Atlantic and Eurasia overlapped with that of EA/WR teleconnection pattern. Therefore, the positive phase of the EA/WR pattern could be stimulated or enhanced by positive $SST_{EP}$ and then lead to weak ventilation conditions that were beneficial for the occurrence of haze. In 2016, the positive $SST_{EP}$ in autumn were

consistent with the positive correlation fields, leading to more $DHD_{NH}$. Furthermore, the $DHD_{NH}$ varied with an obvious decreasing trend from 2006 to 2010, and with a dramatic increasing trend after 2010. The variation of the EA/WR and $SST_{EP}$ index exhibited similar features. During the most recent 10 years, the EA/WR pattern was distributed as its strongest negative phase in 2010 and strongest positive phase in 2016, which was consistent with the variation of $DHD_{NH}$ (Figure 11). The variation of the EA/WR pattern could largely explain the trend break of $DHD_{NH}$. As shown by Gao and Chen (2017),

October SST anomalies near the Gulf of Alaska and the subtropical eastern Pacific contributed to the haze pollution over North China in October 2016. Impacts of the $SST_{EP}$ on ventilation conditions were robust and could continue into December. Furthermore, the relationships with autumn SST in Atlantic were also examined to be weaker and significant only in small regions (figure omitted).

The Eurasian snowpack and atmospheric circulation dominant modes were stably coupled from autumn to the subsequent

spring (Sun 2017), so the role of the preceding October–November (ON) snow cover were also examined (Figure 12). The snow cover over western Siberia ($Snow_{WS}$) was significantly correlated with $DHD_{NH}$ (EA/WR) (Figure 12a), i.e., the correlation coefficient was 0.52 (0.45) after detrending (Table 1). More snow was correlated with a higher albedo, resulting in a colder land surface. When there was higher $Snow_{WS}$, negative Z500 anomalies over western Siberia and positive anomalies over eastern China, i.e., the two active centers of the EA/WR pattern in the east, were significantly stimulated.

The WAF associated with positive $Snow_{WS}$ anomalies was evidently induced near western Siberia and efficiently propagated westwards, and stimulated an obvious anti-cyclone over Baikal Lake and NH area (Figure 12b). The $Snow_{WS}$ varied similarly with $DHD_{NH}$ and achieved its maximum (minimum) in 2016 (2010). As revealed by Wang et al (2015) and Yin et al (2017), the preceding autumn Arctic sea ice has a close relationship with the winter haze days in the east of China. The climatic relationship between the Arctic sea ice and $DHD_{NH}$ and the anomalies in December were also examined and found

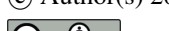



to be not significant (figure omitted). This may be due to the relationship being different in the early and late winter, which

requires more research in the future. The detailed mechanisms between the external forcings and haze pollution should be

studied physically and dynamically in future work. Furthermore, although anthropogenic emissions were limited during haze

pollution events, there were still aerosols being discharged in the atmosphere by the dense population and industry before

and during 16–21 December. There is little doubt that the high concentration of $PM_{2.5}$ was the fundamental cause for haze

pollution, and the associated atmospheric anomalies played key roles in the severe haze pollution events. The previous

accumulation of atmospheric particles also contributed to the occurrence of haze pollution events. As revealed in this study,

it was difficult to modify the simultaneous atmospheric circulations, which significantly contributed to the haze. Therefore,

the controlling measures of anthropogenic emissions should be implemented in advance to reduce the stock of aerosols in the

atmosphere.


**Acknowledgement**

This research was supported by the National Key Research and Development Plan (2016YFA0600703), National Natural

Science Foundation of China (41421004) and the CAS-PKU Partnership Program.

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

**Table and Figure Captions:**

**Table 1** Correlation coefficients between the $DHD_{NH}$ and key indices. The Corr Coe[1] and Corr Coe[2] indicated that correlation coefficients were calculated after and before detrending. The AC was the anticyclone index that was defined as the mean Z500 over 105–125$^o$E, 30–50$^o$E. The local PBL, surface wind speed and relative humidity were calculated as the mean over the NH area. All the correlation coefficients were above the 99% confidence level.

**Figure 1** (a) The variation of $DHD_{NH}$ from 1979 to 2016 and (b) the parameters of the main haze processes in China in 2016: haze (blue bar) and severe haze (red bar) cover area, the maximum $PM_{2.5}$ concentration (yellow line, right y-axis, unit: μg/m$^3$) and the number of lasting days (green line and number).

**Figure 2** Distribution of the global atmospheric circulation anomalies, (a) Z500 (shading) and U200 (contour) and (b) SLP (shading) and SAT (contour) in December 2016. The anomalies here are calculated with respect to the period from 1981–

255    2010.



**Figure 3** Distribution of the regional atmospheric circulation anomalies, (a) the height of PBL (shading), surface lift index (contour) and wind at 850 hPa (arrow); and (b) surface wind speed (shading) and surface relative humidity (contour) in December 2016. The anomalies here are calculated with respect to the period from 1981–2010.

**Figure 4** (a) the measures to limit anthropogenic emissions and (b) the maximum hourly $PM_{2.5}$ concentration during 16–21 December 2016. In panel (a), the red rectangle indicates both the vehicle control and production restriction measures that were implemented, while the blue triangle indicates the production that was restricted. The letters in panel (b) were the names of the provinces.

**Figure 5** The variation of area-mean visibility (black), surface wind speed (blue) and surface relative humidity (red, right y-axis). The intensity of the temperature inversion ($T_{925}$–$T_{1000}$) in Beijing is shown as a gray bar.

**Figure 6** Distribution of the global atmospheric circulation anomalies, (a) Z500 (shading) and U200 (contour); the white dots indicate Z500 anomalies exceeding the 95% confidence level (t test); and (b) SLP (shading) and SAT (contour) during 16–21 December 2016; the white dots indicate SLP anomalies exceeding the 95% confidence level (t test). The anomalies here are calculated with respect to the period of 1981–2010.

**Figure 7** Vertical-latitude section ($110^{o}$-$120^{o}$ E mean) of wind during 16–21 December 2016, omega (shading) and wind (arrow, omega was magnified 100 times).

**Figure 8** The variation of area-mean anomalous height of PBL in December 2016. The anomalies here are twice a day and calculated with respect to the December mean PBLH from 1981 to 2010.

**Figure 9** Distribution of the regional atmospheric circulation anomalies, surface wind (arrow) and surface relative humidity (shading) during 16–21 December 2016. The white dots indicate surface relative humidity anomalies exceeding the 95% confidence level (t test). The anomalies here are calculated with respect to the period from 1981–2010.

**Figure 10** (a) The correlation coefficients (shading) between the preceding autumn SST and $DHD_{NH}$, and the anomalous SST in 2016 (contour) that are calculated with respect to the period from 1979–2016; and (b) The correlation coefficients between $SST_{EP}$ and Z500 exceeding the 90% confidence level (shading), correlated WAF (arrow) and quasi-geostrophic stream function (contour) at 500 hPa in December.





**Figure 11** Variation of the $DHD_{NH}$ (black), EA/WR pattern (red), and $SST_{EP}$ (blue) indices from 1979 to 2016. The solid

lines indicate the indices whose linear trends were removed and the symbols without lines were the original indices.

    **Figure 12** (a) The correlation coefficients (shading) between the October-November snow cover and $DHD_{NH}$. The dots

indicate the correlation coefficients exceeding the 95% confidence level (t-test); and (b) The correlation coefficients between

$snow_{WS}$ and Z500 exceeding the 90% confidence level (shading), correlated WAF (arrow) and quasi-geostrophic stream

function (contour) at 500 hPa in December.

    **Figure 13** Variation of the $DHD_{NH}$ (black) and $Snow_{WS}$ (blue) indices from 1979 to 2016. The solid lines indicate the indices

whose linear trends were removed and the symbols without lines are the original indices.








Table 1 The correlation coefficients between the $DHD_{NH}$ and key indices. The Corr Coe[1] and Corr Coe[2] indicate the correlation coefficients that were calculated after and before detrending. The AC was the anticyclone index that was defined as the mean Z500 over 105–125°E, 30–50°E. The local PBL, surface wind speed and relative humidity were calculated as the mean over the NH area. All the correlation coefficients were above the 99% confidence level.

| Index | EA/WR | AC | PBL | Wind Speed | Humidity | $SST_{EP}$ | $Snow_{WS}$ |
|---|---|---|---|---|---|---|---|
| Corr Coe[1] | 0.66 | 0.62 | −0.59 | −0.63 | 0.49 | 0.55 | 0.52 |
| Corr Coe[2] | 0.66 | 0.62 | −0.54 | −0.62 | 0.46 | 0.54 | 0.50 |

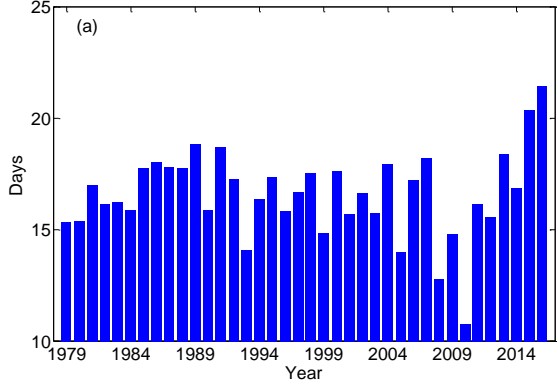

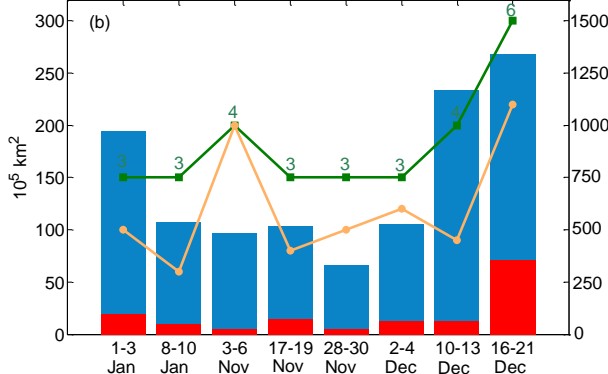

Figure 1 (a) The variation of $DHD_{NH}$ from 1979 to 2016 and (b) the parameters of the main haze processes in China in 2016: haze (blue bar) and severe haze (red bar) cover area, the maximum $PM_{2.5}$ concentration (yellow line, right y-axis, unit: μg/m$^3$) and the number of lasting days (green line and number).



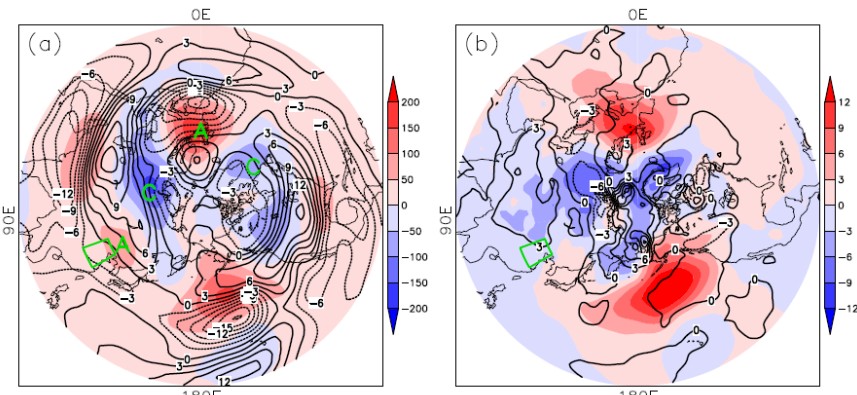

Figure 2 Distribution of the global atmospheric circulation anomalies, (a) Z500 (shading) and U200 (contour) and (b) SLP (shading) and SAT (contour) in December 2016. The anomalies here are calculated with respect to the period from 1981–2010.


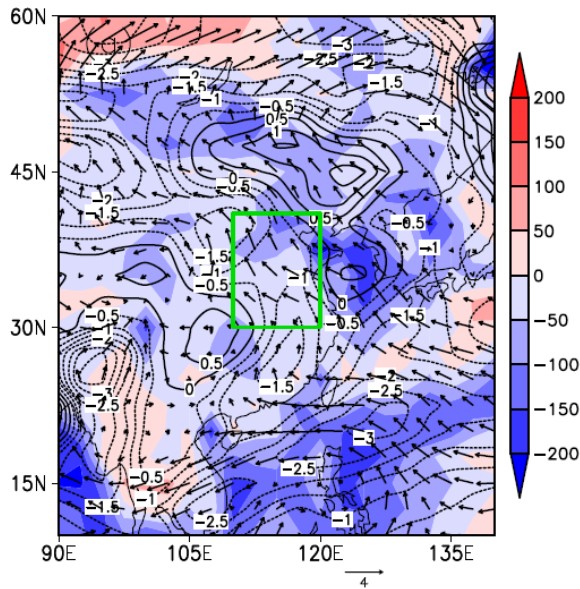





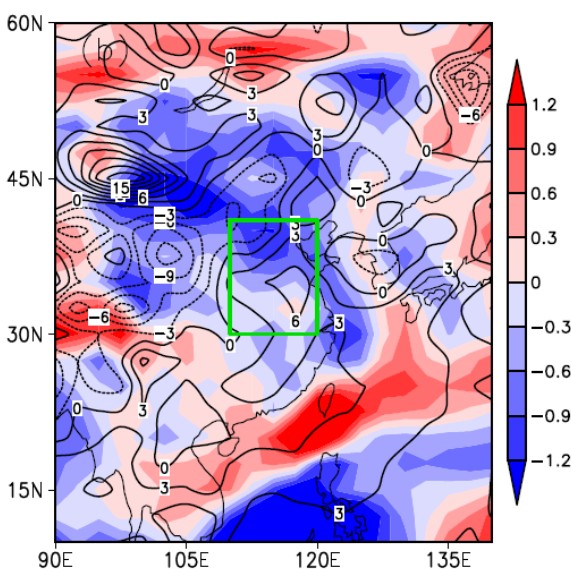

Figure 3 Distribution of the regional atmospheric circulation anomalies, (a) the height of PBL (shading), surface lift index
(contour) and wind at 850 hPa (arrow); and (b) the surface wind speed (shading) and surface relative humidity (contour) in
December 2016. The anomalies here are calculated with respect to the period from 1981–2010.

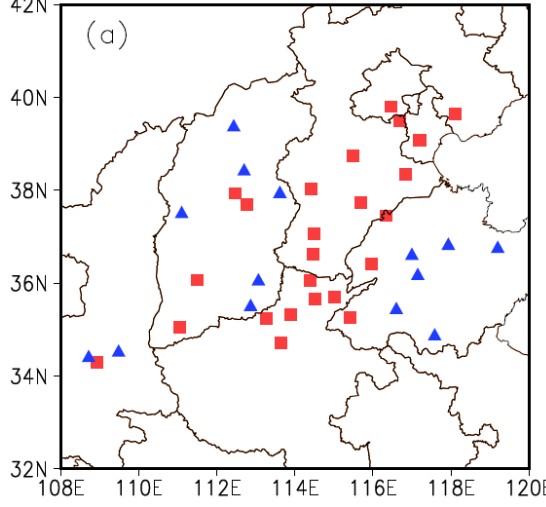



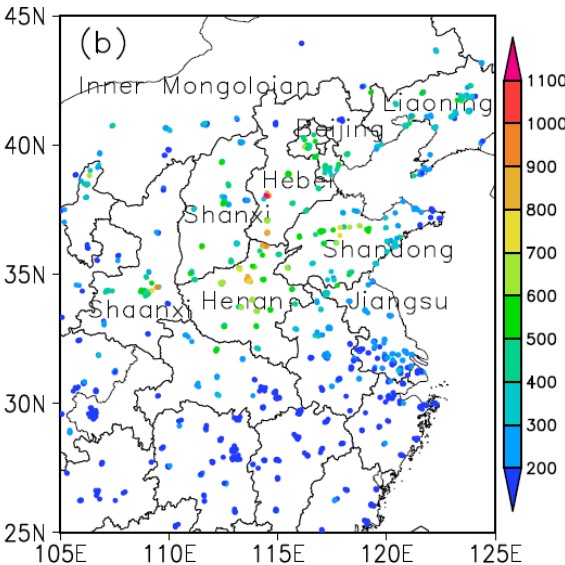

Figure 4 (a) the measures to limit anthropogenic emissions and (b) the maximum hourly PM$_{2.5}$ concentration during the
period of 16–21 December 2016. In panel (a), the red rectangle indicates both the vehicle control and production restriction
measures that were implemented, while the blue triangle indicates the production that was restricted. The letters in panel (b)
were the names of the provinces.

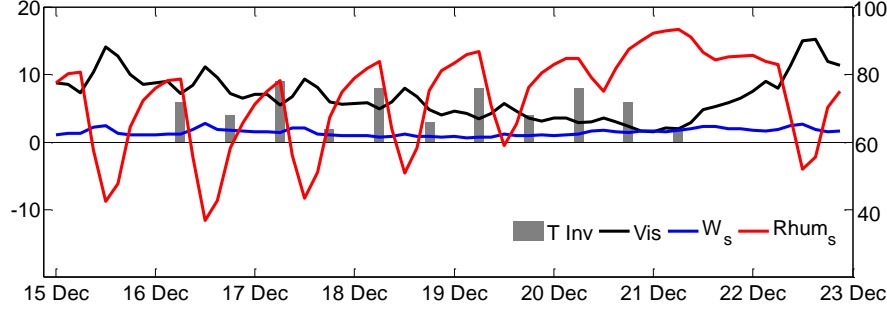

Figure 5 The variation of area-mean visibility (black), surface wind speed (blue) and surface relative humidity (red, right
y-axis). The intensity of the temperature inversion (T$_{925}$–T$_{1000}$) in Beijing is shown as gray bar.




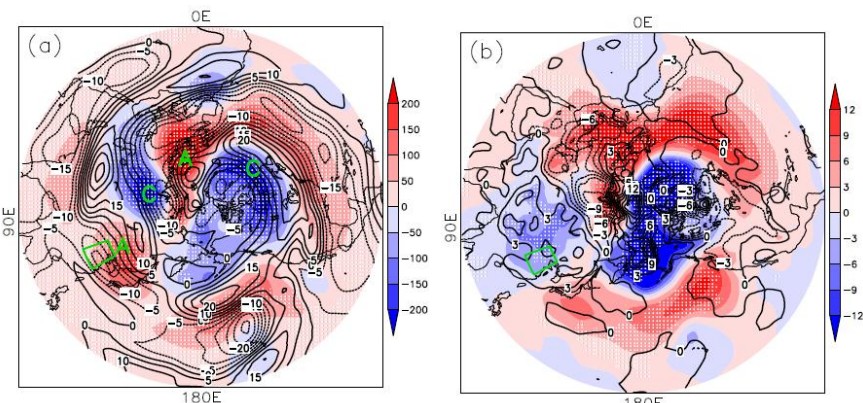

Figure 6 Distribution of the global atmospheric circulation anomalies, (a) Z500 (shading) and U200 (contour); the white dots
indicate Z500 anomalies exceeding the 95% confidence level (t test). (b) SLP (shading) and SAT (contour) in 16–21
December 2016; the white dots indicate SLP anomalies exceeding the 95% confidence level (t test). The anomalies here are
calculated with respect to the period of 1981–2010.

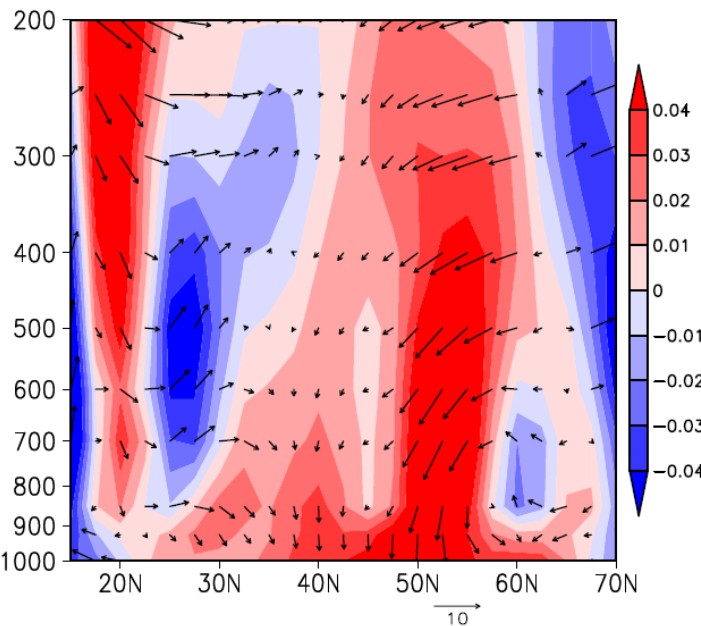

Figure 7 Vertical-latitude section (110°-120°E mean) of wind during 16–21 December 2016, omega (shading) and wind
(arrow, omega was magnified 100 times).



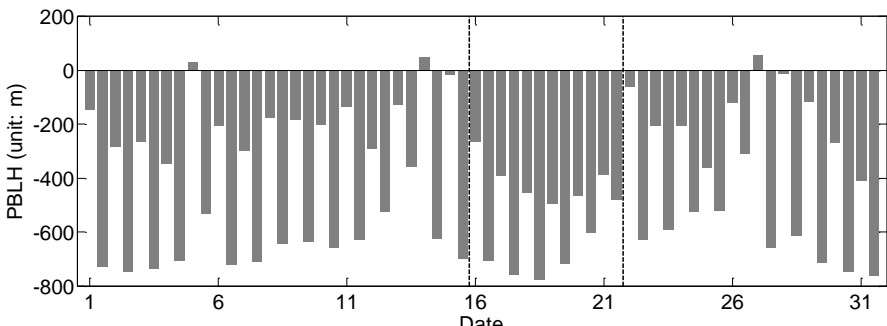

Figure 8 The variation of area-mean anomalous height of PBL in December 2016. The anomalies here are twice a day and calculated with respect to the December mean PBLH from 1981 to 2010.


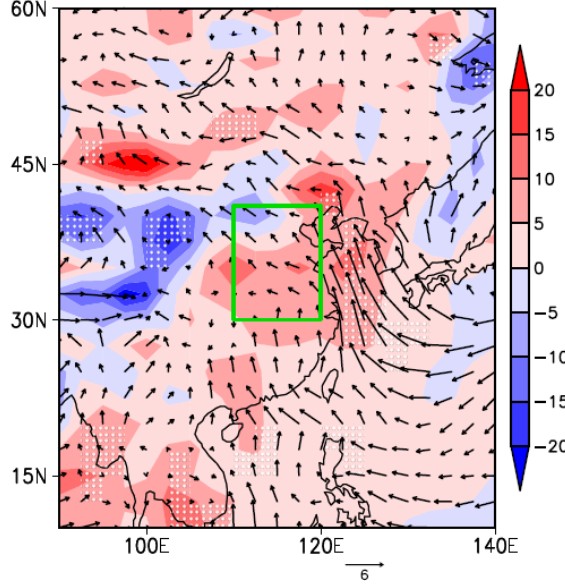

Figure 9 Distribution of the regional atmospheric circulation anomalies, surface wind (arrow) and surface relative humidity (shading) during 16–21 December 2016. The white dots indicate surface relative humidity anomalies exceeding the 95% confidence level (t test). The anomalies here are calculated with respect to the period from 1981–2010.




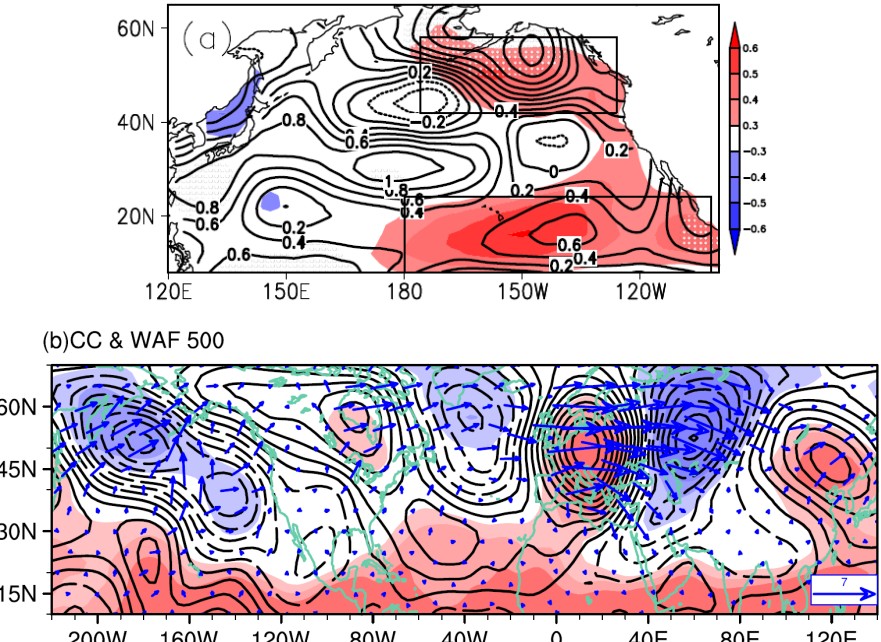

Figure 10 (a) The correlation coefficients (shading) between the preceding autumn SST and $DHD_{NH}$, and the anomalous SST in 2016 (contour) that are calculated with respect to the period from 1979–2016; and (b) The correlation coefficients between

$SST_{EP}$ and Z500 exceeding the 90% confidence level (shading), correlated WAF (arrow) and quasi-geostrophic stream function (contour) at 500 hPa in December.

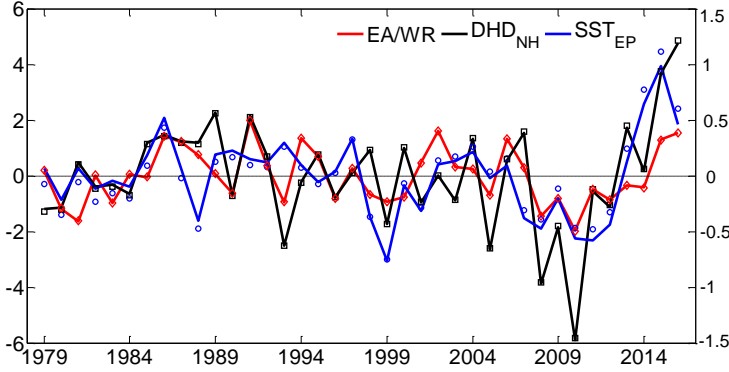

Figure 11 Variation of the $DHD_{NH}$ (black), EA/WR pattern (red), and $SST_{EP}$ (blue) indices from 1979 to 2016. The solid lines indicate the indices whose linear trends were removed and the symbols without lines are the original indices.





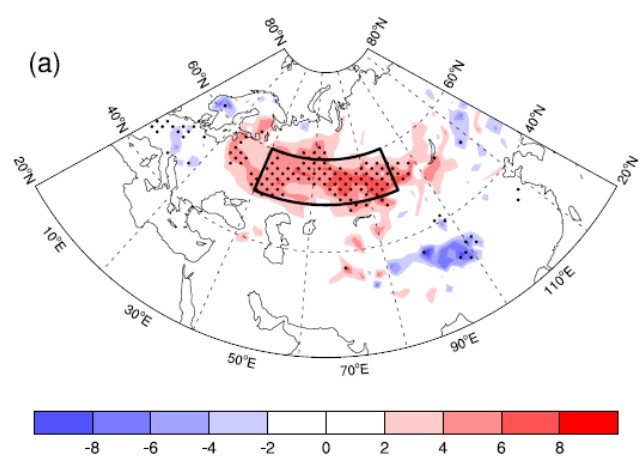


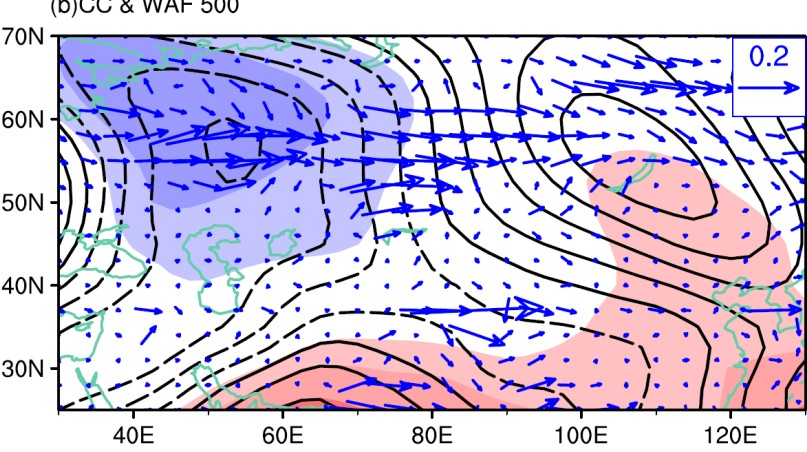

Figure 12 (a) The correlation coefficients (shading) between the October-November snow cover and $DHD_{NH}$. The dots indicate the correlation coefficients exceeding the 95% confidence level (t-test); and (b) The correlation coefficients between $snow_{WS}$ and Z500 exceeding the 90% confidence level (shading), correlated WAF (arrow) and quasi-geostrophic stream

function (contour) at 500 hPa in December.





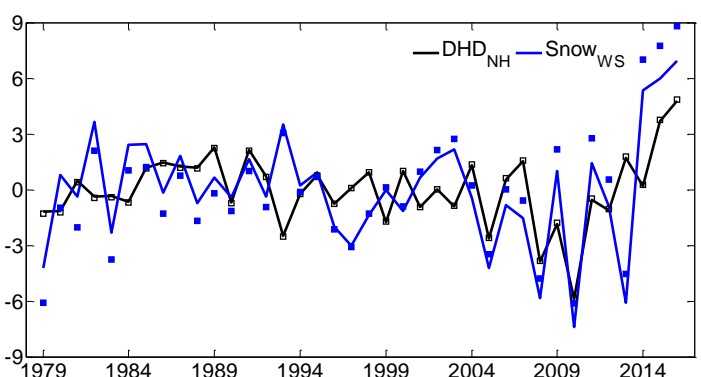

Figure 13 Variation of the $DHD_{NH}$ (black) and $Snow_{WS}$ (blue) indices from 1979 to 2016. The solid lines indicate the indices
whose linear trends were removed, and the symbols without lines are the original indices.