# Peer review of "Role of Atmospheric Circulations on Haze Pollution in December 2016"

_Atmospheric Chemistry and Physics, 2017_

## Referee Comment (RC1) · Anonymous Referee #1 · 9 Jul 2017

Review of "Role of atmospheric circulations on haze pollution in December 2016" by Yin and Wang (MS ID: ACP-2017-521).

In December 2016, the eastern part of China has suffered from the most severe haze pollution during the past decades. This study revealed the possible roles of atmospheric circulations on this haze pollution event in addition to the pollutant emissions, which is interesting and also important to increase understanding the impact of climate change on the increased haze pollutions in China. However, there are still some key points need to be clear in the future. 1. In this study, the authors have indicated that there are several atmospheric circulations including the East Asian jet, EA/WR pattern, PBL, wind speed etc. present significant correlations with the variations of December haze occurrences over North China. Most of these factors have been already revealed

in the earlier papers, also including the works from the authors. However, the possible physical mechanisms for these factor influences have not been well addressed before, also in this paper. So, more discussions on this aspect are suggested in the next version. 2. I think the "Abstract" should be reworded. Just present the correlations between the meteorological factors and haze occurrences is not enough, more discussions related their impacting processes a needed. Also, the key issues related why the haze pollution in December 2016 is the most severe in the past decades and why the event in 16-21 December 2016 is the most severe during December should be clear. 3. Actually, the meteorological conditions for the haze occurrences in December that revealed in this study present similar with that for the events in winter season. There is no new factor has been revealed in this study. The most important thing of this study is to address the reasons for the severest haze pollution in December 2016 and the event in 16-21 December. I think these key issues are not clear at the current MS. 4. what the definition of the "surface lift index"? how to calculate it? 5. why not used NCEP/NCAR II? Or use ERA-Interim dataset across the study? 6. which period is used for the correlation calculation in Table 1? In my first view, I think these correlations are just for the December 2016.

---

## Referee Comment (RC2) · Anonymous Referee #2 · 11 Jul 2017

Air pollution is a complicated question, which is caused by both anthropogenic emissions and meteorological conditions. In this study, the authors investigated the possible mechanism for the several haze in December 2016 from the meteorological aspect. They concluded some atmospheric, oceanic, and snow cover factors which are related to the haze in eastern China. The results are interesting and important for us to understand the variability of haze over eastern China. I recommend the publication of the manuscript. However, the manuscript needs some revisions before it can be considered for publication, which can potentially contribute to enhance the value of the manuscript. Specified comments: (1) The abstract should be reworded to including more physical processes. (2) Datasets and method: what is the variable of "surface"? The NCEP/NCAR data are from 1948 onward, not from 1979. (3) For the reanalysis

data, I recommend the authors use the NCEP/NCAR or ERA-Interim for analysis. Using a dataset across the study can assure the match guarantee the consistency among the variables. (4) What time period for the correlations in Table 1? (5) There is no (a) and (b) in Figure 3. (6) Figure 6 is the averaged mean over 16-21 December 2016? How did the authors calculate the significant test in Figure 6? Similar question is also to Figure 7, 9, and 10. (7) The authors should clearly show the definition for the EA/WR pattern, haze event, and surface lift index.

---

## Author Comment (AC1) · 12 Aug 2017

In December 2016, the eastern part of China has suffered from the most severe haze pollution during the past decades. This study revealed the possible roles of atmospheric circulations on this haze pollution event in addition to the pollutant emissions, which is interesting and also important to increase understanding the impact of climate change on the increased haze pollutions in China. However, there are still some key points need to be clear in the future.

**1. In this study, the authors have indicated that there are several atmospheric circulations including the East Asian jet, EA/WR pattern, PBL, wind speed etc. present significant correlations with the variations of December haze occurrences over North China. Most of these factors have been already revealed in the earlier papers, also including the works from the authors. However, the possible physical mechanisms for these factor influences have not been well addressed before, also in this paper. So, more discussions on this aspect are suggested in the next version.**

**Reply:**

(1) **More discussions about the physical mechanisms have been included in the revision version**. Most of the earlier papers, including the works of the authors, focused on the variations of **winter haze** and associated atmospheric circulations. Differently, the issues addressed by this manuscript were the variation of **December haze and the extreme events in 2016** which was not studied before. Actually, when we analyzed the **sub-seasonal** variation of haze occurrence, we found that the monthly characteristics were different. Thus, we tried to explore the reasons. This manuscript focused on December, and some other works about January and February haze were underway.

(2) **Some comparisons with winter haze were included in Section 3**, such as follows.

"To verify the relationship between the $DHD_{NH}$ and EA/WR pattern, the correlation coefficient was calculated; it was 0.66 from 1979 to 2016 after removing the linear trend and exceeded the 99% confidence level (Table 1). This positive correlation was stronger than that with winter haze days (i.e., 0.43), as analyzed by Yin et al. (2017)."

The main conclusions were that the associated circulations with December haze were similar, but **more significant than those with winter haze**. The underlying indication was that the correlated atmospheric circulations in late winter were possibly different with those associated with winter haze. Furthermore, the **plausible external forcings** were also discussed and the **snow cover**, as a new factor, was analyzed. The significantly correlated regions of SST (i.e. near the Gulf of Alaska and the subtropical eastern Pacific) were also different from that in winter (i.e. the subtropical western Pacific).

**Revision, in section 3:**

Physically, the positive phase of EA/WR pattern strengthened the anomalous

anti-cyclone over NH area and Japan Sea from surface to the middle troposphere, resulting in confined vertical motion. The southerly anomalies made the cold air and surface wind speed weaker, but enhanced the humid flow. Under control of such atmospheric circulations and local meteorological conditions, the horizontal and vertical dispersion of atmospheric particulates was suppressed. Thus, the pollutants gathered within a narrow space and the haze occurred frequently. In addition, high humidity supported a beneficial environment for the hygroscopic growth of haze matters. In December 2016, the height of PBL was the lowest and the intensity of the anomalous anticyclone over the NH was the strongest (Table 1), indicating that the vertical dispersion condition of pollutants was the weakest during the past 38 years. The other key indices (i.e., the EA/WR index, surface wind speed and surface relative humidity) all were in top-six. Thus, the atmospheric circulations and local meteorological conditions were strongly benefit for haze occurrence and combined to result in the severest haze events in December 2016.

**2. I think the "Abstract" should be reworded. Just present the correlations between the meteorological factors and haze occurrences is not enough, more discussions related their impacting processes a needed. Also, the key issues related why the haze pollution in December 2016 is the most severe in the past decades and why the event in 16-21 December 2016 is the most severe during December should be clear.**

**Reply:**

(1) Following the advice, the "Abstract" was reworded to include the physical processed.

(2) In table 1, we added one row to **evaluate the ranks of key indices in 2016**. The atmospheric circulations and local meteorological conditions were strongly benefit for haze occurrence and combined to result in the severest haze events in December 2016. For the synoptic case in 16–21 December, the anthropogenic emissions were strongly confined. The limitation of pollutant emission was strongest in 2016, and this is the reason why we chose the synoptic case. **Under such a background, the severest haze pollution still occurred, which emphasized the role of atmosphere.** The anomalies of atmospheric circulation were similar to those throughout December, but they were **more evident and much stronger than the mean status in December 2016** that resulted in the severest December haze.

**Revision, in the Abstract and Section 3 & 4**

……The atmospheric circulations must play critical roles in the sub-seasonal haze events. Actually, the positive phase of East Atlantic/West Russia pattern in the middle troposphere strengthened the anomalous anti-cyclone over NH area that confined vertical motion below. The associated southerly anomalies made the cold air and surface wind speed weaker, but enhanced the humid flow. Thus, the horizontal and vertical dispersion of atmospheric particulates was suppressed and the pollutants

gathered within a narrow space. In December 2016, these key indices were strongly beneficial for haze occurrence and combined to result in the severest haze pollution. The influences of preceding autumn sea surface temperature near the Gulf of Alaska and the subtropical eastern Pacific, October-November snow cover in western Siberia and associated physical processes on haze pollution were also discussed.

……In December 2016, the height of PBL was the lowest and the intensity of the anomalous anticyclone over the NH was the strongest (Table 1), indicating that the vertical dispersion condition of pollutants was the weakest during the past 38 years. The other key indices (i.e., the EA/WR index, surface wind speed and surface relative humidity) all were in top-six. Thus, the atmospheric circulations and local meteorological conditions were strongly benefit for haze occurrence and combined to result in the severest haze events in December 2016.

The anomalies of atmospheric circulation during 16–21 December were similar to those throughout December, but they were more evident and much stronger than the mean status in December 2016 that resulted in the severest December haze…….

Table 1 The correlation coefficients between the $DHD_{NH}$ and key indices from 1979 to 2016 and the ranks of key indices in 2016. The Corr Coe[1] and Corr Coe[2] indicate the correlation coefficients that were calculated after and before detrending. The AC was the anticyclone index that was defined as the mean Z500 over 105–125°E, 30–50°E. The local PBL, surface wind speed and relative humidity were calculated as the mean over the NH area. All the correlation coefficients were above the 99% confidence level. The rank was sorted from largest to smallest, when the Corr Coe was positive. If the Corr Coe was negative, the rank was calculated from smallest to largest.

| Index | EA/WR | AC | PBL | Wind Speed | Humidity | $SST_{EP}$ | $Snow_{WS}$ |
|---|---|---|---|---|---|---|---|
| Corr Coe[1] | 0.66 | 0.62 | −0.59 | −0.63 | 0.49 | 0.55 | 0.52 |
| Corr Coe[2] | 0.66 | 0.62 | −0.54 | −0.62 | 0.46 | 0.54 | 0.50 |
| Rank | 3 | 1 | 1 | 6 | 6 | 4 | 1 |

**3. Actually, the meteorological conditions for the haze occurrences in December that revealed in this study present similar with that for the events in winter season. There is no new factor has been revealed in this study. The most important thing of this study is to address the reasons for the severest haze pollution in December 2016 and the event in 16-21 December. I think these key issues are not clear at the current MS.**

**Reply:**

 (1) The issues addressed by this manuscript were the variation of **December haze and the extreme events in 2016** which was not studied in the earlier papers. **Some comparisons with winter haze were included in Section 3**. The main conclusions were that the associated circulations with December haze were similar, but **more significant than those with winter haze**. Furthermore, the **plausible external forcings** were also discussed and the October-November **snow cover**, as a new factor,

was analyzed. The significantly **correlated regions of SST** (i.e. near the Gulf of Alaska and the subtropical eastern Pacific) were also **different** from that in winter (i.e. the subtropical western Pacific).

(2) In table 1, we added one row to **evaluate the ranks of key indices in 2016**. The atmospheric circulations and local meteorological conditions were **strongly benefit for haze occurrence and combined to result in** the severest haze events in December 2016. For the synoptic case in 16–21 December, the anthropogenic emissions were strongly confined. The limitation of pollutant emission was strongest in 2016, and this is the reason why we chose the synoptic case. **Under such a background, the severest haze pollution still occurred, which emphasized the role of atmosphere.** The anomalies of atmospheric circulation were similar to those throughout December, but they were **more evident and much stronger than the mean status in December 2016** that resulted in the severest December haze.

**Revision:**

……In December 2016, the height of PBL was the lowest and the intensity of the anomalous anticyclone over the NH was the strongest (Table 1), indicating that the vertical dispersion condition of pollutants was the weakest during the past 38 years. The other key indices (i.e., the EA/WR index, surface wind speed and surface relative humidity) all were in top-six. Thus, the atmospheric circulations and local meteorological conditions were strongly benefit for haze occurrence and combined to result in the severest haze events in December 2016.

The anomalies of atmospheric circulation during 16–21 December were similar to those throughout December, but they were more evident and much stronger than the mean status in December 2016 that resulted in the severest December haze…….

Table 1 The correlation coefficients between the $DHD_{NH}$ and key indices from 1979 to 2016 and the ranks of key indices in 2016. The Corr Coe[1] and Corr Coe[2] indicate the correlation coefficients that were calculated after and before detrending. The AC was the anticyclone index that was defined as the mean Z500 over 105–125°E, 30–50°E. The local PBL, surface wind speed and relative humidity were calculated as the mean over the NH area. All the correlation coefficients were above the 99% confidence level. The rank was sorted from largest to smallest, when the Corr Coe was positive. If the Corr Coe was negative, the rank was calculated from smallest to largest.

| Index | EA/WR | AC | PBL | Wind Speed | Humidity | $SST_{EP}$ | $Snow_{WS}$ |
|---|---|---|---|---|---|---|---|
| Corr Coe[1] | 0.66 | 0.62 | −0.59 | −0.63 | 0.49 | 0.55 | 0.52 |
| Corr Coe[2] | 0.66 | 0.62 | −0.54 | −0.62 | 0.46 | 0.54 | 0.50 |
| Rank | 3 | 1 | 1 | 6 | 6 | 4 | 1 |

**4. what the definition of the "surface lift index"? how to calculate it?**

**Reply:**

(1) Surface lifted index is the lifted index at 500-mb based on the surface parcel and

was calculated as the difference between the temperature of the parcel when it was lifted to the upper level and the surrounding temperature. The surface lifted index is a measurement of the stability of an air mass at a given moment.

(2) The indicative function of surface lifted index was repeated with PBLH and anti-cyclone anomalies. **To make the Figure 3(a) clearer, the surface lifted index was unused in the revision version.**

**Revision:**

[Figure]

Figure 3 Distribution of the regional atmospheric circulation anomalies, (a) the height of PBL (shading), and wind at 850 hPa (arrow);

**5. why not used NCEP/NCAR II? Or use ERA-Interim dataset across the study?**

**Reply:**

Almost all of the datasets was downloaded from NCEP/NCAR across the study. However, the website of NCEP/NCAR did not support the height of PBL. Thus, only the PBLH was derived from ERA-Interim and the reason was explained the Section 2.

**Revision, in section 2:**

……For the representativeness of vertical dispersion, the $1^{o} \times 1^{\circ}$ height of PBL (not available on the website of NCEP/NCAR) was also used here, but derived from the ERA-Interim dataset (Dee et al. 2011). ……

**6. which period is used for the correlation calculation in Table 1? In my first view, I think these correlations are just for the December 2016.**

**Reply:**

The period was from 1979 to 2016.

**Revision:**

Table 1 The correlation coefficients between the $DHD_{NH}$ and key indices from 1979 to 2016 and the ranks of key indices in 2016. The Corr Coe¹ and Corr Coe² indicate the correlation coefficients that were calculated after and before detrending. The AC was the anticyclone index that was defined as the mean Z500 over 105–125°E, 30–50°E. The local PBL, surface wind speed and relative humidity were calculated as the mean over the NH area. All the correlation coefficients were above the 99% confidence level. The rank was sorted from largest to smallest, when the Corr Coe was positive. If the Corr Coe was negative, the rank was calculated from smallest to largest.

| Index | EA/WR | AC | PBL | Wind Speed | Humidity | $SST_{EP}$ | $Snow_{WS}$ |
|---|---|---|---|---|---|---|---|
| Corr Coe¹ | 0.66 | 0.62 | −0.59 | −0.63 | 0.49 | 0.55 | 0.52 |
| Corr Coe² | 0.66 | 0.62 | −0.54 | −0.62 | 0.46 | 0.54 | 0.50 |
| Rank | 3 | 1 | 1 | 6 | 6 | 4 | 1 |

---

## Author Comment (AC2) · 12 Aug 2017

Air pollution is a complicated question, which is caused by both anthropogenic emissions and meteorological conditions. In this study, the authors investigated the possible mechanism for the several haze in December 2016 from the meteorological aspect. They concluded some atmospheric, oceanic, and snow cover factors which are related to the haze in eastern China. The results are interesting and important for us to understand the variability of haze over eastern China. I recommend the publication of the manuscript. However, the manuscript needs some revisions before it can be considered for publication, which can potentially contribute to enhance the value of the manuscript.

Specified comments:

**1. The abstract should be reworded to including more physical processes.**

**Reply:**

Following the advice, the "Abstract" was reworded to include the physical processed.

**Revision, in the Abstract**

……The atmospheric circulations must play critical roles in the sub-seasonal haze events. Actually, the positive phase of East Atlantic/West Russia pattern in the middle troposphere strengthened the anomalous anti-cyclone over NH area that confined vertical motion below. The associated southerly anomalies made the cold air and surface wind speed weaker, but enhanced the humid flow. Thus, the horizontal and vertical dispersion of atmospheric particulates was suppressed and the pollutants gathered within a narrow space. In December 2016, these key indices were strongly beneficial for haze occurrence and combined to result in the severest haze pollution. The influences of preceding autumn sea surface temperature near the Gulf of Alaska and the subtropical eastern Pacific, October-November snow cover in western Siberia and associated physical processes on haze pollution were also discussed.

**2. Datasets and method: what is the variable of "surface"? The NCEP/NCAR data are from 1948 onward, not from 1979.**

**Reply:**

The variable on surface is wind.

The available period of NCEP/NCAR data was changed to "from 1948 to 2016".

**Revision, in section 2:**

The geopotential height at 500 hPa (Z500), zonal wind at 200 hPa (U200), wind at 850 hPa, wind at surface, sea level pressure (SLP), surface air temperature (SAT), surface relative humidity and vertical wind (omega) were available on the website of the National Centers for Environmental Prediction/National Center for Atmospheric Research (NCEP/NCAR). These NCEP/NCAR reanalysis I datasets had a horizontal resolution of 2.5 °×2.5 °from 1948 to 2016 (Kalnay et al. 1996).

**3. For the reanalysis data, I recommend the authors use the NCEP/NCAR or ERA-Interim for analysis. Using a dataset across the study can assure the match guarantee the consistency among the variables.**

**Reply:**

Almost all of the datasets was downloaded from NCEP/NCAR across the study. However, the website of NCEP/NCAR did not support the height of PBL. Thus, only the PBLH was derived from ERA-Interim and the reason was explained the Section 2.

**Revision, in section 2:**

……For the representativeness of vertical dispersion, the $1°\times1°$ height of PBL (not available on the website of NCEP/NCAR) was also used here, but derived from the ERA-Interim dataset (Dee et al. 2011). ……

**4. What time period for the correlations in Table 1?**

**Reply:**

The period was from 1979 to 2016.

**Revision:**

Table 1 The correlation coefficients between the $DHD_{NH}$ and key indices from 1979 to 2016 and the ranks of key indices in 2016. The Corr Coe[1] and Corr Coe[2] indicate the correlation coefficients that were calculated after and before detrending. The AC was the anticyclone index that was defined as the mean Z500 over 105–125°E, 30–50°E. The local PBL, surface wind speed and relative humidity were calculated as the mean over the NH area. All the correlation coefficients were above the 99% confidence level. The rank was sorted from largest to smallest, when the Corr Coe was positive. If the Corr Coe was negative, the rank was calculated from smallest to largest.

| Index | EA/WR | AC | PBL | Wind Speed | Humidity | SST$_{EP}$ | Snow$_{WS}$ |
|---|---|---|---|---|---|---|---|
| Corr Coe[1] | 0.66 | 0.62 | −0.59 | −0.63 | 0.49 | 0.55 | 0.52 |
| Corr Coe[2] | 0.66 | 0.62 | −0.54 | −0.62 | 0.46 | 0.54 | 0.50 |
| Rank | 3 | 1 | 1 | 6 | 6 | 4 | 1 |

**5. There is no (a) and (b) in Figure 3.**

**Reply:**

The (a) and (b) was not clear in the submitted Figure 3 and they were revised now.

**Revision:**

[Figure]

Figure 3 Distribution of the regional atmospheric circulation anomalies, (a) the height of PBL (shading) and wind at 850 hPa (arrow); and (b) the surface wind speed (shading) and surface relative humidity (contour) in December 2016. The anomalies here are calculated with respect to the period from 1981–2010.

**6. Figure 6 is the averaged mean over 16-21 December 2016? How did the authors calculate the significant test in Figure 6? Similar question is also to Figure 7, 9, and 10.**

**Reply:**

The data to plotted Figure 6, 7 and 9 was daily reanalysis and long-term-mean daily reanalysis. Thus, the significant test was calculated **basing on the daily data** to evaluate whether the atmospheric circulations in 16–21 December 2016 was significantly different from the climate mean status.

**7. The authors should clearly show the definition for the EA/WR pattern, haze event, and surface lift index.**

**Reply:**

(1) The definition of EA/WR pattern was clearly showed in the revised versiom..

(2) The definition and calculation process of haze event was revised and briefly introduced. The full calculation method was well introduced in the referred work of the authors, i.e. Yin et al. 2017. If we repeated the similar content here which is not necessary, the manuscript would become too long.

(3) **To make the Figure 3(a) clearer, the surface lifted index was unused in the revision version.** The indicative function of surface lifted index was repeated with PBLH and anti-cyclone anomalies. The definition of surface lifted index was also

showed below.

Surface lifted index is the lifted index at 500-mb based on the surface parcel and was calculated as the difference between the temperature of the parcel when it was lifted to the upper level and the surrounding temperature. The surface lifted index is a measurement of the stability of an air mass at a given moment.

**Revision:**

(1) ……The EA/WR pattern consisted of four anomalous centers and the positive phase is associated with positive anomalous height over Europe and northern China, and negative anomalies over the central North Atlantic and north of the Caspian Sea. The EA/WR index was computed by the NOAA climate prediction center according to the Rotated Principal Component Analysis used by Barnston et al. (1987)…….

(2) ……The routine meteorological measurements included relative humidity, visibility and wind speed at surface that were collected eight times per day. The temperature profile was collected with a sounding balloon twice per day. The calculation procedure for the haze data was consistent with that of Yin et al. (2017), which was mainly based on the observed visibility and relative humidity…….

(3)

[Figure]

Figure 3 Distribution of the regional atmospheric circulation anomalies, (a) the height of PBL (shading), and wind at 850 hPa (arrow);